

# Longitudinal analysis of step counts in Parkinson's disease patients: insights from a web-based application and generalized additive model

Yuan Gu[1] and Yishu Gong[2]

[1] School of Medicine, Stanford University, Palo Alto, CA, United States of America
[2] The Harvard T.H. Chan School of Public Health, Harvard University, Boston, MA, United States of America

## ABSTRACT

**Background.** Parkinson's disease (PD) is a chronic neurological disorder that affects millions of people worldwide. A common motor symptom associated with PD is gait impairment, leading to reduced step count and mobility.

**Methods.** Monitoring and analyzing step count data can provide valuable insights into the progression of the disease and the effectiveness of various treatments. In our study, the generalized additive model (GAM) was used to identify statistically significant variables for step counts. Additionally, a web application was developed as an interactive visualization tool.

**Results.** The GAM model shows that the following variables are statistically significant for daily step counts: sex ($p = 0.03$), handedness ($p = 0.015$), PD status of father ($p = 0.056$), COVID-19 status (Yes *vs.* No, $p = 0.008$), cohort (PD *vs.* Healthy, $p < 0.0001$), the cubic regression spline with three basis functions of age by cohorts ($p < 0.0001$), and the random effect of individual age trajectories ($p = 0.0001$).

**Conclusions.** Based on the PPMI data, we find that sex, handedness, PD status of father, COVID-19 status, cohort, and the smoothing functions of age are all statistically significant for step counts. Additionally, a web application tailored specifically for step count analysis in PD patients was developed. This tool provides a user-friendly interface for patients, caregivers, and healthcare professionals to track and analyze step count data, facilitating personalized treatment plans and enhancing the management of PD.

# INTRODUCTION

Parkinson's disease (PD) has emerged as the second most prevalent age-related neurodegenerative disorder worldwide, resulting in a substantial burden of disability and a significantly increased risk of developing dementia and mortality. The overall prevalence rate of PD is 572 cases per 100,000 among individuals aged 45 and above, and over 93 cases per 100,000 among individuals aged 65 and older (*Willis et al., 2022*). Despite significant progress in PD medications over the past half-century, none can be classified as a cure. Although these drugs have improved in effectiveness and duration, they still do not fully alleviate PD symptoms or offer a long-term solution. Therefore, treatment options must be

Corresponding author
Yishu Gong,
yishugong@hsph.harvard.edu

customized to meet the specific needs of each individual patient (*Jankovic & Aguilar, 2008*; *Ellis & Fell, 2017*; *Stoker & Barker, 2020*). Additionally, the cause of PD remains unknown, and there are no established methods to prevent the disease. While several risk factors, such as pesticide exposure, have been identified, confirmed causes of PD are genetic in nature. When PD is not linked to genetics, it is categorized as idiopathic (iPD), indicating that the precise cause remains unclear (*Mahlknecht et al., 2022*; *Puigròs et al., 2022*).

Today, the emergence of wearable solutions—such as smartwatches, fitness trackers, and health monitoring devices—has become increasingly popular due to their ability to provide low-cost, instant, convenient, and accurate measurements. These wearables offer a wide range of functions, including tracking physical activity, monitoring vital signs, and even detecting certain health conditions. With their affordability, real-time data availability, and ease of use, wearable solutions have empowered individuals to take charge of their health and well-being in an accessible and accurate manner (*Channa, Popescu & Ciobanu, 2020*).

Thanks to the reliability of wearable solutions, monitoring physical activity, especially during mild to moderate PD stages, holds particular significance. Previous research has extensively investigated clinically significant daily physical activities, providing evidence related to physical performance metrics such as gait, muscle strength, and step counts. Individuals with PD typically exhibit lower levels of physical activity compared to individuals of similar age without the condition. Even those newly diagnosed and not yet on anti-Parkinson medications typically take around 5,000 steps per day if they are community-dwelling and able to walk independently. This falls short of the recommended 7,000 steps per day for older adults to maintain good health, highlighting the sedentary lifestyles often observed in people with PD (*von Rosen et al., 2021*).

Therefore, step counts over time serve as a crucial indicator of movement in telemonitoring, with previous studies consistently demonstrating that continuous monitoring of step counts may be an important way to quantify declining ambulatory behavior due to disease progression or improved ambulatory behavior resulting from rehabilitation, medical, or surgical interventions in persons with PD (*Cavanaugh et al., 2012*; *Master et al., 2022*). These findings provide valuable insights for informing public health recommendations on physical activity, aiming to enhance prevention strategies for better overall health outcomes (*Gong et al., 2023*).

## RELATED WORK

As discussed in the introduction, PD has a high prevalence, particularly among the elderly, with no current medications that offer a cure. Step counts serve as a valuable indicator of health and well-being in PD patients. Additionally, there is a strong correlation between reduced step counts and increased mortality rates among PD patients, as demonstrated in multiple studies. One study (*Stens et al., 2023*) analyzed 111,309 individuals across 12 studies and found that significant risk reductions in all-cause mortality were observed at 2,517 steps per day. Another study (*Inoue et al., 2023*) of 3,101 adults found a curvilinear dose–response relationship, showing that taking 8,000+ steps on multiple days per week lowered the 10-year risk of all-cause and cardiovascular mortality. Other studies, such

as *Banach et al. (2023)*, *Paluch et al. (2022)* and *Ahmadi et al. (2024)*, also indicated that a similar daily step count can reduce mortality risk, even in individuals with high sedentary time (*Stens et al., 2023*).

Several studies have focused on PD patients. For example, the WATCH-PD study assessed whether a combination of a smartwatch and smartphone could accurately measure early PD-related features (*Adams et al., 2023*). The Personalized Parkinson's Project (PPP), a single-center study of patients with early-stage PD in the Netherlands co-sponsored by Verily (Google Life Sciences), aimed to more precisely measure aspects of patient function and well-being (*Diao et al., 2022*). Another study investigated predictors of sustained physical activity in PD patients during the COVID-19 pandemic in Sweden, using step count data from the Actigraph GT3x, although it included only 63 subjects with a one-year follow-up (*Moulaee Conradsson et al., 2024*). In 2024, the WATCH-PD team published their 12-month findings in *npj Parkinson's Disease* (*Adams et al., 2023*). This study explored gait and tremor metrics from commercially available smartwatches and smartphones to assess treatment efficacy for early-stage PD. The study enrolled eighty-two individuals with early, untreated PD and fifty age-matched controls across seventeen US research sites. The authors emphasized the importance of step count as a critical indicator of patient burden and mortality. Over 12 months, they observed a mean daily step count decrease among PD patients not taking dopaminergic medication. However, these studies face limitations due to small sample sizes and brief follow-up durations.

A landmark study is the Parkinson Progression Marker Initiative (PPMI), an ongoing longitudinal study conducted by The Michael J. Fox Foundation. It aims to identify biomarkers for PD progression, improving disease understanding and enhancing therapeutic trials targeting PD modification (*Marek et al., 2011*). In 2018, Verily partnered with PPMI to use a multi-sensor investigational wearable device for continuous data collection on movement, physiological measures, and environmental factors (*Atri et al., 2022*).

Other studies, such as the BETA-PD project (*von Rosen et al., 2021*), aim to delineate physical activity (PA) profiles—namely "Sedentary," "Light Movers," and "Steady Movers"—of 301 individuals with PD to yield valuable insights for customizing PA interventions. *De Carvalho Lana et al. (2021)* investigated the validity of various mHealth devices (Google Fit, Health, STEPZ, Pacer, Fitbit) in estimating step counts among 34 individuals with idiopathic PD. Despite the growing interest in PD research, few studies have analyzed step count trends over extended periods across various age groups for both healthy individuals and PD patients. Although some studies using PPMI data have examined features such as brain changes (*Koros et al., 2023*), blood markers of inflammation (*Bartl et al., 2023*), and MDS-UPDRS Scores (*Holden et al., 2018*), minimal research has focused on physical activity. Studies (*Tsukita, Sakamaki-Tsukita & Takahashi, 2022*; *Amara et al., 2019*) examined self-reported activity using the Physical Activity Scale for the Elderly, but neither analyzed objective step count data from PPMI, possibly due to the large dataset from years of passive monitoring.

Moreover, large sample sizes in this area of research are lacking. The absence of an open-source web application also limits clinicians' ability to examine step count trends

over time among different groups, based on factors like sex, COVID-19 status, and hand dominance. Without such a tool, the exploration of potential correlations and insights that could contribute to personalized treatment approaches and further understanding of the disease is restricted.

Therefore, it is essential to conduct a comprehensive longitudinal analysis of step counts, incorporating potential risk factors, using extensive, robust, long-term follow-up data. The PPMI study, for example, offers a rich data source well-suited to such analyses, enabling more definitive conclusions about the implications of step count variations over time.

Wearable technology has ranked as the top fitness trend for the past two years, and forecasted financial trends suggest its use will continue to grow annually in the foreseeable future. One study (*Montes et al., 2020*) evaluated step-count functionality in wearable devices, providing strong evidence of their reliability and validity for tracking steps during free movement. To date, to the best of our knowledge, features (*e.g.*, gait, speech) were extracted using the same methods described in the baseline manuscript, with the exception of step count and a composite speech measure, which had not been previously analyzed (*Adams et al., 2023*).

Walking is the most common form of human movement, with a safe and efficient gait essential for maintaining independence throughout life. Gait is a key indicator of overall health, highlighting the importance of regular gait assessments in healthcare settings. However, gait is a complex, multidimensional activity that cannot be fully captured by a single parameter. Methods such as video motion capture systems, force platforms, and instrumented walkways are considered the gold standards for quantitative gait analysis. Although highly accurate, these methods are expensive, resource-intensive, and limited to stationary use in laboratory environments, which restricts their accessibility and practicality for widespread clinical use (*Werner et al., 2023*). Step count analysis using wearable devices offers a more affordable, user-friendly alternative to traditional gait analysis, enabling out-of-lab monitoring without the need for specialized equipment, trained personnel, or controlled conditions. These devices support unobtrusive, continuous tracking over long periods, capturing a more comprehensive view of natural gait and activity in daily life (*Werner et al., 2023*; *Gauthier-Beaupré & Grosjean, 2023*).

Although a few Shiny apps have been published online in our previous studies (*Ahmadi et al., 2024*; *Adams et al., 2023*; *Gu et al., 2023a*; *Gu et al., 2023b*), focusing on different diseases, such as cancer and kidney failure, no accessible online platform currently exists for describing baseline characteristics of PD patients or for modeling time-varying trends in step counts between PD patients and healthy participants. An online app offers greater convenience for clinicians and researchers, allowing real-time step count analysis from their own devices. This study addresses this gap by constructing a longitudinal model based on PPMI data and providing an online tool for clinical investigators to visualize these trends in detail. Our online tool is available at the following link: https://baran-shad.shinyapps.io/Rshiny_PPMIstepcounts/. Portions of this text were previously published as part of a preprint (*Gong et al., 2023*).

## METHODS

We requested access to the full breadth of individual-level PPMI longitudinal data from the Parkinson's Progression Markers Initiative. Details of the PPMI study can be found at the following link: https://www.ppmi-info.org/. A total of 353 subjects recorded their step counts, with a follow-up period exceeding 2.11 years. The cumulative number of steps taken by each subject was recorded on an hourly basis. Our analysis includes additional demographic and characteristic factors, such as sex, handedness (right, left/both), presence of Parkinson's disease in parents (yes or no), COVID-19 status, and cohort (PD or healthy control). After incorporating these variables and excluding subjects who were not successfully enrolled or resided outside the United States during the study, our final sample consists of 172 subjects.

### Matching

To retain as many observations as possible, we employed a 1:4 (32 control *vs.* 126 PD) matching approach to select subjects from the control and PD groups, utilizing the "matchit" package in R. The matching model was specifically designed using the nearest neighbors approach, matching the two groups based on factors such as age, sex, handedness, and COVID-19 status to align their propensity scores. The aim of this 1:4 matching was to pair each treated unit with the four closest control units based on their respective propensity scores. By implementing this matching methodology, we sought to enhance comparability between the treated and control groups, thereby improving the validity of our analysis and mitigating potential confounding factors. After merging with data on the presence of Parkinson's disease in parents, the final analysis dataset contains 32 healthy controls and 126 PD patients.

### Time binning

To account for the varying decimal values of ages represented in days, we employed the binning method in our study. This approach allowed for the creation of equally spaced time bins, specifically using intervals of 3 months. For example, age groups ranging from 66 to 67 were binned into intervals of 0.25. This resulted in age bin labels such as 66.25, 66.50, 66.75, and 67. By grouping the data in this manner, we were able to increase the sample size within each binned age interval and aggregate and cumulate the step counts within each age bin. Using binning in conjunction with cumulative step counts enabled us to capture and analyze overall activity patterns across different age intervals in a more manageable and informative way. This approach also facilitated the examination of age-related patterns and trends with greater resolution, providing valuable insights into the relationship between age and other variables under investigation.

### Baseline characteristics and descriptive plots

There are a total of 32 healthy control subjects and 126 PD patients (Table 1). Among these subjects, 61 are females and 97 are males. In previous studies (*Yust-Katz et al., 2008*; *Van der Hoorn et al., 2012*), left-handedness has been considered a potential risk factor for PD, so we incorporated handedness as a factor in our analysis. Our dataset includes 131

**Table 1  Baseline characteristics by healthy and PD groups.**

|  | Healthy control (n = 32) | Parkinson's disease (n = 126) | P-value |
|---|---|---|---|
| COVID-19 status |  |  | 0.14 |
| No | 23 (72%) | 105 (83%) |  |
| Yes | 9 (28%) | 21 (17%) |  |
| Sex |  |  | 0.8854 |
| Female | 12 (37%) | 49 (39%) |  |
| Male | 20 (63%) | 77 (61%) |  |
| Handedness |  |  | 0.1831 |
| Right | 24 (75%) | 107 (85%) |  |
| Left/Both | 8 (25%) | 19 (15%) |  |
| PD status of father |  |  | 0.0278 |
| No | 32 (100%) | 109 (87%) |  |
| Yes | 0 (0%) | 17 (13%) |  |
| PD status of mother |  |  | 0.0831 |
| No | 32 (100%) | 115 (91%) |  |
| Yes | 0 (0%) | 11 (9%) |  |

right-handed individuals and 27 left-handed individuals or those exhibiting ambidextrous tendencies. The study coincided with the outbreak of the COVID-19 pandemic, which poses significant risks, particularly for the elderly population, including individuals already diagnosed with Parkinson's disease. COVID-19 status was also collected, and there are 30 subjects who had COVID-19. Another unconventional risk factor considered is the PD status of both parents. A total of 17 patients has fathers with Parkinson's disease, and 11 patients have mothers with Parkinson's disease.

The $p$-values for the chi-square test are presented in Table 1 to determine whether there is a statistically significant difference between the two groups (healthy and PD) across different variables. Notably, most of these $p$-values are greater than 0.05, indicating no statistical significance. However, the variable "PD status of Father" has a $p$-value of 0.0278, indicating a significant difference in the PD status of fathers between the healthy and PD groups. For other baseline characteristics, all the $p$-values are > 0.05, suggesting that there is no significant difference in these characteristics between the PD and healthy groups, and we can assume that these variables are evenly distributed between the two groups.

In Fig. 1, the step counts over time are depicted for each subject, with the left side representing the healthy group and the right side illustrating the PD group. Moving on to Fig. 2, it showcases the change in step counts from baseline over time at a 3 months time intervals. The left side pertains to the healthy group (green box), while the right side corresponds to the PD group (gray box). In this figure, each pair of box plots represents the average daily steps over 3-month intervals for two groups. The seven pairs of box plots correspond to baseline (0 months) and subsequent time points at 3, 6, 9, 12, 15, and 18 months. Each box plot displays the distribution of step counts within each group at that

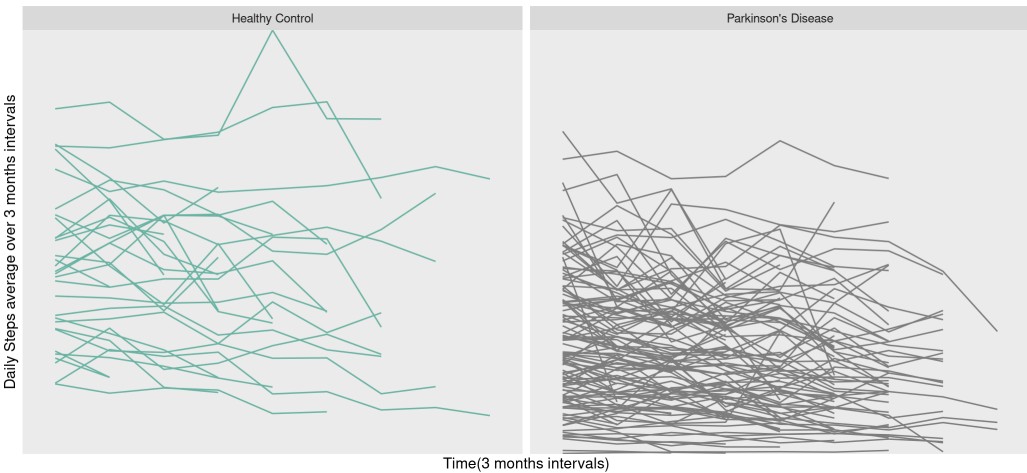

**Figure 1  Trajectories of daily steps over time.**

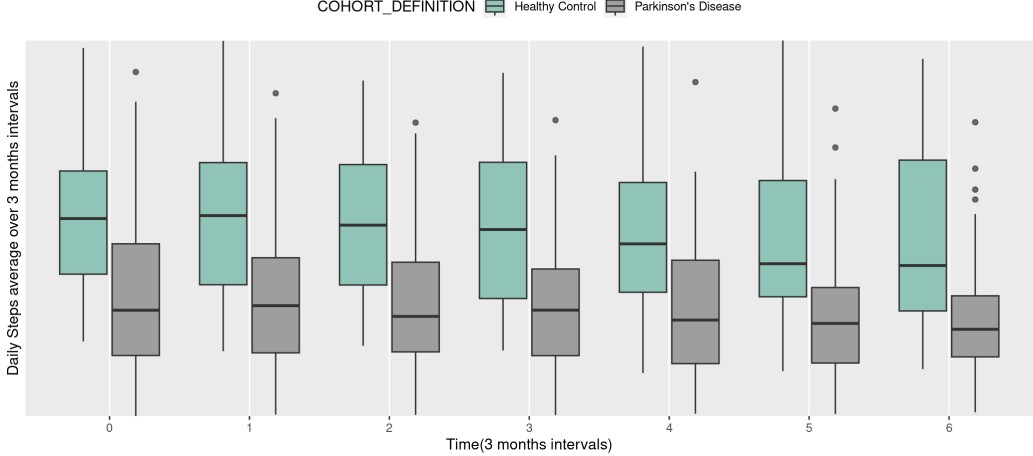

**Figure 2  Box plots of daily steps average over 3 months by different time (3 months intervals).**

specific interval, with the central line indicating the median step count and the height of the box representing the interquartile range, or the spread of middle values.

Overall, the healthy control group consistently shows higher median step counts and broader ranges across all time points compared to the PD group, indicating that healthy individuals maintain greater physical activity levels over time. Meanwhile, the PD group displays lower and relatively stable step counts, suggesting that mobility in PD patients is persistently limited, with minimal variation across the 18-month period. This trend highlights a clear and sustained difference in daily physical activity between the two groups, underscoring the impact of PD on long-term mobility. When comparing the two groups in Fig. 1, it becomes evident that individuals in the healthy group take more daily steps

compared to PD patients. Furthermore, we observe greater variability in step counts among PD patients over time, suggesting a potential decline in daily step counts as PD progresses.

Overall, in Fig. 2, these box plots clearly illustrate the difference in step counts between the two groups at every time point. The trend of higher step counts among individuals in the healthy group, as opposed to the PD group, is consistently depicted in both Figs. 1 and 2. Additional descriptive plots are available in our application for more comprehensive insights.

## GENERALIZED ADDITIVE MODEL RESULTS

The standard linear mixed-effects model is a common choice for longitudinal data analysis (*Hanff et al., 2024*). However, it may not effectively capture the curvature of time-varying trajectories, especially when quadratic relationships exist in repeated measurements. To address this limitation, we employed the generalized additive model (GAM) in our study. Unlike the standard linear mixed-effects model, which relies solely on linear feature contributions, GAM is constructed as a sum of smooth functions of features. This flexibility allows GAM to better accommodate and model complex, non-linear relationships within the data, enhancing the accuracy and interpretability of our analysis.

Here, we employ a GAM similar to the one used in *Adams et al. (2023)*. This choice addresses the challenges of handling step count data, which tends to be dense, irregularly sampled, and highly variable across individuals due to differing personal habits. While the WATCH-PD study collected limited step count data, they were unable to apply the framework to step count data and reported only empirical results. In our analysis of the richer PPMI step count data, we extend the model by incorporating an additional two years of step count tracking and integrating multiple covariates.

Our specific model is defined as follows:

$$steps_{i,j} = \beta_0 + \beta_1 * Sex + \beta_2 * Handedness + \beta_3 * PD_{father} + \beta_4 * PD_{mother} + \beta_5 * Covid$$
$$+ \beta_6 * Cohort + \beta_7 f_1(age_i, cohort) + \beta_8 f_{2j}(age_i, subject_j) + \varepsilon_{ij}.$$

In this model, step count serves as the response variable. The independent variables include sex, handedness, parental PD status (father and mother), COVID-19 status, and cohort (Parkinson's Disease *vs.* Healthy Control). We applied a cubic regression spline with three basis functions to model age within cohorts, as well as random effects to capture individual age trajectories. The temporal variable is modeled as a piecewise cubic Hermite interpolating polynomial (PCHIP) spline with four basis functions, allowing us to estimate distinct smooth functions across time (*Eilers & Marx, 1996*). For random effects, we introduced a random intercept and smooth temporal trajectories for each participant, accounting for participant-specific variation.

For complex models such as GAMs, standard goodness-of-fit metrics can be inadequate because the non-linear, smooth terms cannot be effectively summarized by simple linear correlations. Therefore, we employed residual diagnostics to assess the model's fit more

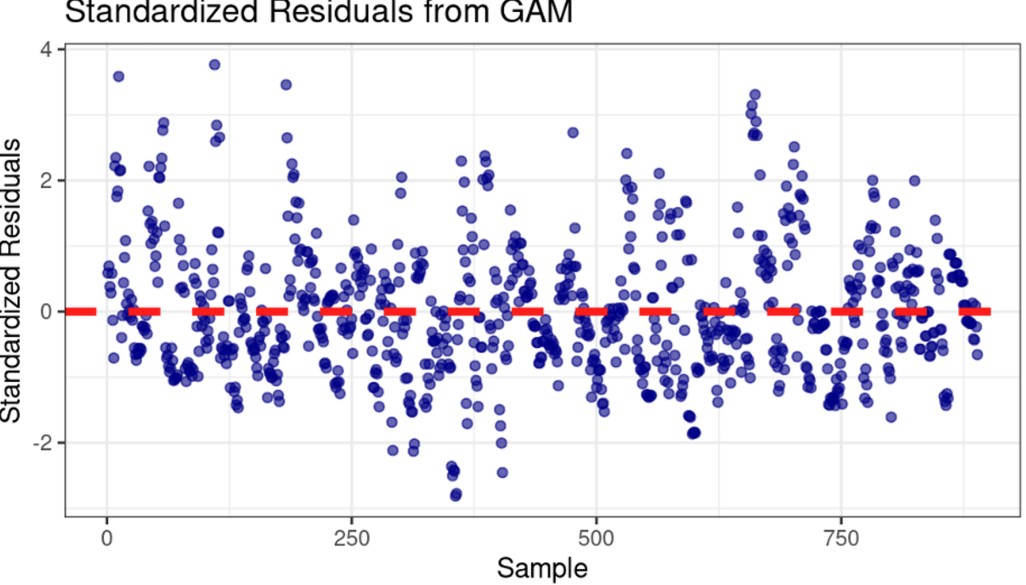

**Figure 3  Standardized residuals *vs.* sample for GAM model diagnostics.**

appropriately. As shown in Fig. 3, the residuals appear normally distributed and centered around zero, suggesting that the model captures the main structure of the data well.

The Q-Q plot of standardized residuals (Fig. 4) specifically examines the normality assumption. The plot shows that residuals near the center closely follow the theoretical quantile line, indicating that the model accurately represents central trends. However, a rightward deviation in the upper tail suggests a slight positive skew, likely influenced by extreme values from individual events, such as travel or other variations from normal activity.

The model result is displayed in Table 2. We observe that several variables, including sex ($p$-value = 0.03), handedness ($p$-value = 0.015), PD status of the father ($p$-value = 0.056, close to the significance level), COVID-19 status ($p$-value = 0.008), cohort ($p$-value < 0.0001), as well as the smoothing curve on age ($p$-value < 0.0001), and the random effect of the smoothing trajectory on age ($p$-value =0.0001), all demonstrate statistical significance or are close to significance in relation to step counts, except for the PD status of the mother ($p$-value = 0.47). Notably, among these features, handedness displays a positive coefficient, implying that right-handed individuals exhibit higher step counts compared to their left-handed counterparts or those who are ambidextrous. Furthermore, as the study extended beyond the COVID-19 period, it's noteworthy that COVID-19 status is another factor that statistically affects daily step counts.

Figure 5 below depicts the step counts estimated by our GAM model. These results are presented as smooth curves, with green curves representing the healthy group and gray curves for the PD group. Additionally, the original daily step data is overlaid as connected lines, providing background information to enhance the clarity and comprehensibility of the visualization. This visualization reinforces the observation that individuals within
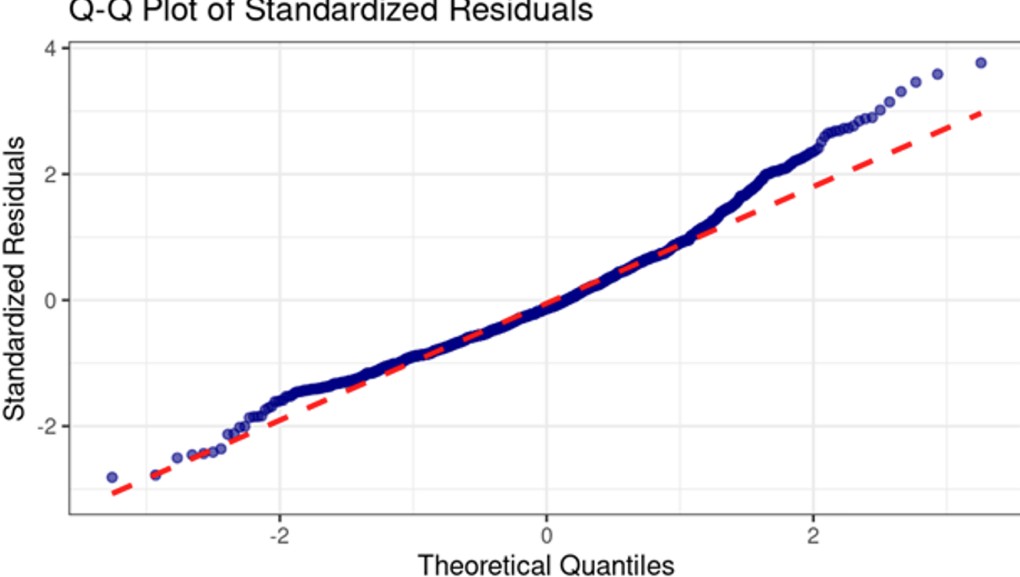

**Figure 4 Q-Q plot of standardized residuals for GAM model diagnostics.**

**Table 2 GAM model results.**

| Variable | Coefficient | *p*-value |
| --- | --- | --- |
| Sex | 388.8 | 0.03035 |
| Handedness (Yes: Right; No; Left/Both) | 564.5 | 0.01545 |
| PD status of father | 588.4 | 0.05633 |
| PD status of mother | −251.9 | 0.47156 |
| COVID-19 | 538.5 | 0.00871 |
| Cohort (PD: 1; Healthy control: 2) | 1,906.6 | <0.0001 |
| Smoothing on age | – | <0.0001 |
| Random effect of smoothing trajectory on age | – | 0.0001 |

the PD group exhibit significantly fewer step counts compared to the control group. Moreover, in both groups, step counts decrease with age, especially after age 70. The figures are displayed within the Shiny application, allowing clinicians to customize the presentation of the figures by making selections from a drop-down menu.

## DISCUSSION

Although numerous studies have concentrated on PD, there is a notable scarcity of investigations that delve into step counts of PD patients over extended periods. This limitation is primarily due to the large volume of daily data records required, making continuous follow-up challenging. Even in the extensive PPMI study, the number of participants in the healthy control group followed for up to two years remains limited compared to the PD groups, leading to imbalanced data between the two groups. This imbalance may constrain the model's ability to offer generalized conclusions. However,

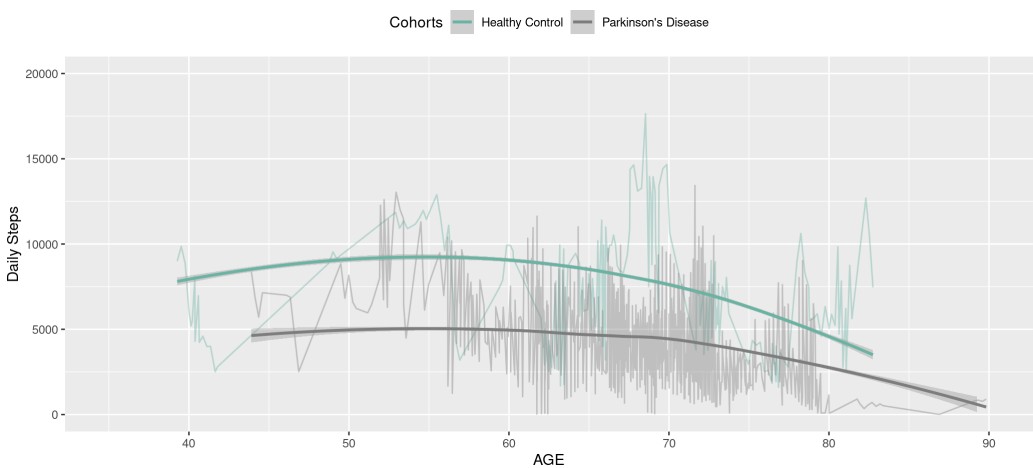

**Figure 5** Daily steps comparison between healthy (blue) and PD groups over age curves: GAM fitted curve; connected lines: original daily steps.

our analysis effectively handled the voluminous and complex daily step count data derived from the PPMI.

Throughout our study, several statistically significant factors affecting step counts were identified. These factors include gender, COVID-19 status, handedness, the PD status of the father, as well as the application of cubic spline regression for age and the inclusion of individual age as a random effect.

The GAM model revealed that, after adjusting for other variables, handedness carries a positive coefficient, suggesting that right-handed individuals record higher step counts than left-handed or ambidextrous individuals. This finding aligns with previous studies and may serve as a foundation for future research examining the imaging disparities between right- and left-handed individuals. Potential analysis methods could include exploring time-course differences in functional loci between these groups through advanced imaging techniques, such as image segmentation using location networks combined with 3D U-net or augmented pyramid networks like QAP-NET and RAR-U-Net, *etc* (*Wang et al., 2023*; *Wang & Voiculescu, 2021*; *Wang, Zhang & Voiculescu, 2021*; *Song, Gu & Kumar, 2023*; *Gu et al., 2024*).

Gender also plays a significant role in step counts, with men exhibiting higher daily step counts compared to women. The curvature of cubic regression over age is another significant factor, as daily step counts tend to decrease with age, particularly after the age of 70. Furthermore, COVID-19 status has proven to be a statistically significant factor impacting daily step counts. Another noteworthy finding pertains to the family history of PD: while the PD status of the father approached statistical significance ($p$-value = 0.056), the small subset of patients with a parental history of PD limits our ability to fully explore familial influences. Additional data collection is necessary to conduct a more comprehensive investigation of this factor.

The flexibility of the GAM model is one of its strengths, allowing for the integration of varying degrees of complexity within the data. By adjusting smoothness degrees, GAM

can effectively capture both linear and polynomial patterns. Moreover, GAMs inherently account for autocorrelation in longitudinal data, unlike linear mixed models, which rely on a predefined correlation matrix that may misrepresent the actual data structure. Additionally, the polynomial smoothing method in GAMs can automatically interpolate missing data, providing a more reliable approximation compared to traditional methods that assume specific missing data patterns (*e.g.*, missing completely at random or missing at random). However, the use of cubic regression splines as the basis for the GAM model adds complexity to model interpretation due to the flexibility of basis functions. While this enhances the model's ability to capture intricate patterns, it may make interpreting individual effects more challenging. Nevertheless, our goal is to identify statistically significant risk factors for step counts, and GAM proves to be an appropriate approach for this purpose.

To further enhance accessibility and facilitate deeper analysis, we developed an R Shiny web application. This tool allows investigators to interactively explore descriptive plots, visualize step count trajectories over time for different groups, and understand the model results directly through a user-friendly interface. The app provides clinicians and researchers with the flexibility to examine demographic insights and group key features for a more comprehensive understanding of the statistical significance of various risk factors on step counts in the context of Parkinson's Disease.

## CONCLUSION

Overall, our research builds upon our previously published study (*Marek et al., 2011*) and addresses a significant gap in the literature, as limited studies have focused on step counts—an essential metric for tracking PD patients. Step count data has been largely overlooked due to several challenges, including small sample sizes, short follow-up periods, complex study designs, and the practical difficulties associated with ensuring consistent daily use of digital wearables by PD patients.

We utilized the Parkinson's PPMI data, which includes longitudinal data on hundreds of patients followed for over two years, capturing a wide range of important features. By applying an advanced GAM, we were able to accommodate polynomial data structures and address missing data through smoothing interpolation. This approach is a significant improvement over the commonly used linear mixed models, which only account for linear relationships and rely on a predefined correlation matrix that often fails to accurately represent the data's true correlation structure.

Our findings are groundbreaking in identifying innovative features such as handedness and the PD status of the father as statistically significant factors affecting step counts. These factors have never been identified in similar studies before, adding valuable insights to the field of PD research. Additionally, we developed a free web application that allows users to visualize step count trajectories over time across different groups, offering a dynamic and user-friendly tool for enhanced data analysis and interpretation. This app enables investigators to conveniently access and interpret the model results directly *via* a shared link.

Despite the advancements in our study, several limitations should be acknowledged. The sample size of the control group is relatively modest, which could potentially constrain the robustness of comparative analyses. Additionally, the number of independent variables considered in our study is limited, and there may be other factors contributing to step counts that were not explored. To address these limitations and gain a more comprehensive understanding of the dynamics of PD progression, a larger and more diverse dataset, coupled with an extended follow-up period, is necessary. This would not only provide a broader context for the analysis but also offer the opportunity to explore the nuances of PD progression and its interactions with various factors in greater depth.

## ACKNOWLEDGEMENTS

We like to thank all patients and healthy volunteers for participating in the PPMI study, all investigators who contributed, and the Michael J. Fox Foundation.

### Funding
The authors received no funding for this work.

### Competing Interests
The authors declare there are no competing interests.

### Author Contributions
- Yuan Gu performed the experiments, analyzed the data, prepared figures and/or tables, and approved the final draft.
- Yishu Gong conceived and designed the experiments, analyzed the data, authored or reviewed drafts of the article, and approved the final draft.

### Human Ethics
The following information was supplied relating to ethical approvals (i.e., approving body and any reference numbers):

The data is requested from the PPMI study, the PPMI sites received approval from an ethical standards committee on human experimentation before study initiation and obtained written informed consent for research from all participants in the study.

### Data Availability
The raw data is available from the Parkinson's Progression Markers Initiative. Investigators seeking access to PPMI data are required to sign the Data Use agreement and submit an application: https://www.ppmi-info.org/access-data-specimens/download-data.

### Supplemental Information
Supplemental information for this article can be found online at http://dx.doi.org/10.7717/peerj.19519#supplemental-information.

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
