# Peer review of "Longitudinal analysis of step counts in Parkinson’s disease patients: insights from a web-based application and generalized additive model"

_PeerJ, doi:10.7717/peerj.19519_

## Round 0.1 · original submission · Major Revisions

Dear Co-Authors:

Thank you for submitting your paper to PeerJ Journals. After reviewing it, we will consider "Major Reviews" to improve your manuscript.

Regards.

Dr. Manuel Jiménez

Reviewer 1 ·

Basic reporting

1. The authors may need to add more references in the introduction section.
2. The authors may need to add a discussion section.

Experimental design

no comment

Validity of the findings

no comment

Additional comments

1. The authors may need to add more references in the introduction, such as in line 51.
2. The authors need to go through the manuscript carefully and revise the grammar errors, such as line 79 (extra space), line 80 (missing comma before and), line 115 (vs.).
3. The reviewer suggests to avoid using citation number to start sentence, line 82.
4. Line 171, which three figures?
5. In figure 2, what are the seven pairs bar plots meaning?
6. There are some discussions after conclusion, which is not normal. The authors may add a discussion section before conclusion.

Reviewer 2 ·

Basic reporting

The most important issue is the inclusion of citations for data and previous research (lines 28-29, 50-51, 54-56). Citations are crucial for supporting claims, demonstrating the validity of the data, and situating the research within the context of existing literature. Additionally, maintaining consistency in citation format throughout the manuscript is essential. A minor grammatical correction is needed in line 82, where a comma should be used after "manner" instead of a period.

Experimental design

The most significant aspect of your paper is the application of the Generalized Additive Model (GAM). To strengthen the justification for choosing this model, it is recommended to provide a more detailed discussion of its advantages and disadvantages.

Validity of the findings

I commend the authors' innovative approach to researching step counts related to Parkinson's disease. The web application effectively tracks and analyzes step count data, which is crucial for the management of Parkinson's disease. It would be beneficial if the authors could also include a discussion of the current limitations in Parkinson's disease management to further contextualize the significance of their work.

·

Basic reporting

Manuscript ID 102838v1
This paper is related to reviewing the manuscript titled "Longitudinal analysis of step counts in Parkinson's Disease patients: insights from a Web-Based application and generalized additive model"
The study uses the generalized additive model (GAM) to analyze step count data from Parkinson's disease, a chronic neurological disorder, to understand disease progression and treatment effectiveness, and develops an interactive visualization tool for this analysis.
Firstly, Although the proposed study is successful in terms of good performance analysis and evaluation results, organization, presentation, content and results are poor of the paper. So, major revision given in the following items need to be performed.

Experimental design

1) Improve the conclusion section, enhance the manuscript to convey the purpose, objectives, method and major findings, especially results in the items of convenience, interest, comfort, enhancing student’s self-confidence and subjective initiative.
2) Use abbreviations after the first use in the text, in the abstract and throughout the paper.
3) The literature review is quite insufficient in the introduction section. Complete the introduction and literature sections of the manuscript by providing similar studies from the years 2023-2024 and/or new and current studies that will attract the attention of the readers.
4) Neither the mathematical nor algorithmic expressions of these methods are given in the paper text. The authors urgently need to find a solution to this issue, and the mathematical equations of the methods must be given in the paper.
5) What are the contributions of the authors in this study in terms of Web-Based application and generalized additive model? It is essential to clarify this issue.
6) In addition, the proposed model should be compared with new methods, from the results except some figures.
7) Performance analyses and results are very few and insufficient. Increasing the results and including more detailed analyses in the paper would increase the value and scope of this paper.
8) The interpretation of the results and the discussion section are insufficient. These sections should definitely be increased and improved.
9) The conclusion section really needs to be improved
10) The resolution of the figures giving the analysis results should be increased.
11) Clean the paper of English spelling and punctuation errors

Validity of the findings

As above

Additional comments

My decision is major revision. I would like to inform you that if all the requested items are not completed in this revision, my decision will be to reject the application in the second round. Otherwise, I do not see any harm in publishing the manuscript once the above revisions are made.
Best regards.

---

## Round 0.2 · Minor Revisions

Dear Authors:

Thank you for your effort to improve the manuscript to be published on PeerJ, but some minor changes must be made before it can be published. Please attend to reviewer comments.

Best regards.

Dr. Manuel Jimenez

Reviewer 1 ·

Basic reporting

no comment

Experimental design

no comment

Validity of the findings

no comment

Reviewer 2 ·

Basic reporting

I appreciate the author's efforts in improving the references and grammar. However, please double-check the section numbers for accuracy. Regarding citation 1, it primarily focuses on the situation in North America but does not extensively discuss or provide detailed insights into the global context or worldwide implications of the findings.

Experimental design

The explanation of the Generalized Additive Model (GAM) is clear; however, the author only compared it with the standard linear model. What about the mixed-effects model, which was mentioned in the conclusion? It is recommended that a brief comparison with alternative models, such as the mixed-effects model, be included to strengthen the methodological justification. This would help readers better understand why GAM is the most appropriate choice for this study. (As reference: Hanff, A. M., Krüger, R., McCrum, C., Ley, C., & NCER-PD (2024). Mixed effects models but not t-tests or linear regression detect progression of apathy in Parkinson's disease over seven years in a cohort: a comparative analysis. BMC medical research methodology, 24(1), 183. https://doi.org/10.1186/s12874-024-02301-7)

Validity of the findings

The conclusions are generally well-supported by the results. The study excellently presents an interesting approach to analyzing step counts in Parkinson's Disease (PD) patients using a generalized additive model (GAM) and a web-based application. The use of PPMI data adds clinical significance to the findings, making the results more meaningful for practical applications.

---

## Round 0.3 · accepted · Accept

Dear Authors:

It is a pleasure to inform you that your manuscript, "Longitudinal analysis of step counts in Parkinson's Disease patients: insights from a Web-Based application and generalised additive model", has been accepted for publication.

Congratulations, and thank you for thinking of Peerj journals.

Dr. Manuel Jiménez

Reviewer 2 ·

Basic reporting

no comment

Experimental design

no comment

Validity of the findings

no comment

Additional comments

no comment